# Controllable protein design via autoregressive direct coupling analysis conditioned on principal components

Francesco Caredda[1]*, Lisa Gennai[2], Paolo De Los Rios[2,3], Andrea Pagnani[1,4,5]

**1** Department of Applied Science and Technology, Politecnico di Torino, Torino, Italy, **2** Institute of Physics, School of Basic Sciences, École Polytechnique Fédérale de Lausanne - EPFL, Lausanne, Switzerland, **3** Institute of Bioengineering, School of Basic Sciences, École Polytechnique Fédérale de Lausanne - EPFL, Lausanne, Switzerland, **4** Italian Institute for Genomic Medicine, IRCCS Candiolo, Candiolo, Italy, **5** INFN, Sezione di Torino, Torino, Italy

* francesco.caredda@polito.it

## Abstract

We present FeatureDCA, a statistical framework for protein sequence modeling and generation that extends Direct Coupling Analysis (DCA) with biologically meaningful conditioning. The method can leverage different kinds of information, such as phylogeny, optimal growth temperature, enzymatic activity or, as in the case presented here, principal components derived from multiple sequence alignments, and use it to improve the learning process and consequently efficiently condition the generative process. FeatureDCA allows sampling to be guided toward specific regions of sequence space while maintaining the efficiency and interpretability of Potts-based inference. Across multiple protein families, our autoregressive implementation of FeatureDCA matches or surpasses the generative accuracy of established models in reproducing higher-order sequence statistics while preserving substantial sequence diversity. Structural validation with AlphaFold and ESMFold confirms that generated sequences adopt folds consistent with their intended wild-type targets. In a detailed case study of the Response Regulator family (PF00072), which comprises distinct structural subclasses linked to different DNA-binding domains, FeatureDCA accurately reproduces class-specific architectures when conditioned on subtype-specific principal components, highlighting its potential for fine-grained structural control. Predictions of experimental deep mutational scanning data show accuracy comparable to that of unconditioned autoregressive Potts models, indicating that FeatureDCA also captures local functional constraints. These results position FeatureDCA as a flexible and transparent approach for targeted sequence generation, bridging statistical fidelity, structural realism, and interpretability in protein design.

**Data availability statement:** All datasets used in this work are publicly available at: https://github.com/francescocaredda/FeatureDCAData. The full Julia implementation of `FeatureDCA`, including example Jupyter notebooks for reproducibility and application, can be found at: https://github.com/francescocaredda/FeatureDCA.jl. These repositories provide all the necessary resources to replicate the experiments and analyses described in this study.

**Funding:** FC and AP acknowledge financial support from the project "Explainable Models for Protein Design", funded by the MIUR Progetti di Ricerca di Rilevante Interesse Nazionale (PRIN) Bando 2022 - grant 2022TE5B7X. FC and AP also acknowledge "Centro Nazionale di Ricerca in High-Performance Computing, Big Data and Quantum Computing" (ICSC). This study was carried out within the "FAIR - Future Artificial Intelligence Research" project, and received funding from the European Union NextGenerationEU (Piano Nazionale di Ripresa e Resilienza (PNRR)–Missione 4 Componente 2, Investimento Grants No. 1.3–D.D. 1555 11/10/2022, and No. PE00000013). FC and AP acknowledge support from the European REA, Marie Skłodowska-Curie Actions, grant agreement no. 101131463 (SIMBAD). This paper reflects only the authors' views and opinions, neither the European Union nor the European Commission can be considered responsible for them. LG and PDLR thank the Swiss National Science Foundation for financial support under grant IC00I0-227688. The funders had no role in study design, data collection and analysis, decision to publish, or preparation of the manuscript.

**Competing interests:** The authors have declared that no competing interests exist.

## Author summary

Designing new proteins with desired functions is a major challenge in biology and biotechnology. Current statistical approaches can generate protein-like sequences, but they rarely allow users to guide designs toward specific structures or functions in a clear, interpretable way. We introduce FeatureDCA, a statistical model that learns from sets of evolutionarily related protein sequences how structural and functional properties encoded in the sequences relate to their positions in a low-dimensional projection of protein space. This makes it possible to guide the generation of new protein sequences towards biologically meaningful features. We show that FeatureDCA can generate novel sequences that resemble natural proteins in specific user-defined positions in protein space. In a case study, the method generated bacterial signaling proteins that, when folded with AlphaFold, adopted different dimerization modes highly consistent with experimental structures. FeatureDCA thus provides a transparent and efficient framework for guiding protein design, with potential applications in biotechnology, synthetic biology, and evolutionary research.

## Introduction

The ability to generate novel functional protein sequences is a central challenge in computational biology and protein design. During the past decade, statistical models trained on evolutionary data, particularly those derived from multiple sequence alignments (MSAs), have demonstrated remarkable success in capturing the statistical, structural, and functional constraints that shape natural protein families. Among these, models based on the Potts framework, such as Boltzmann machine Direct Coupling Analysis (bmDCA) [1–4], autoregressive DCA (ArDCA) [5] and more recent latent-variable models such as GENERALIST [6], have established themselves as powerful tools for inferring residue-residue interactions, predicting mutational effects, and designing synthetic sequences with natural-like properties [7].

Despite their success, a fundamental limitation of these models is that, under standard unconditional sampling, the generated sequences primarily reflect the global statistical properties of the training MSA, offering little to no ability to steer generation toward user-defined features or functional subtypes. In practice, however, protein design often demands precisely this kind of control, whether to target a specific structural class, functional behavior, or region in sequence space.

Building on advances in sequence modeling, several generative approaches have emerged in recent years for protein sequence design, including variational autoencoders (VAEs) [8,9], generative adversarial networks (GANs) [10], protein language models (PLMs) [11–18], and diffusion-based methods [19]. Among these, DeepSequence [8] was an early VAE model that demonstrated strong performance in mutational effect prediction and unsupervised modeling of protein families. More recent protein language models leverage large-scale transformer architectures trained on millions of unaligned sequences, either to learn general-purpose protein representations, as in ESM [12,20,21] and ProGen [11], or to improve sequence fitness

prediction by incorporating information from multiple sequence alignments, as in Tranception [22] and PoET-2 [23]. These models have shown impressive capabilities in sequence generation, mutational scanning, and even zero-shot structure prediction. While flexible and expressive, such approaches typically require massive datasets and substantial computational resources, often involving hundreds of millions to billions of parameters, and they generally lack interpretability and direct access to the coevolutionary statistics encoded in MSAs. Moreover, their mechanisms for guiding sequence generation, such as prompt tuning or template-based conditioning, are often indirect or non-transparent, and do not solve the fundamental problem of explicitly conditioning generation on user-defined features within a statistical framework. In contrast, DCA-based architectures are deliberately shallow models that operate directly on aligned homologous sequences, offering a principled, data-efficient, and interpretable alternative that tightly reflects evolutionary constraints and which, with the right extensions, can be made fully controllable.

In this work, we introduce FeatureDCA, a principled extension of the DCA framework that enables conditional generation of protein sequences. Our approach incorporates low-dimensional, biologically meaningful features as conditioning inputs to the model. For the present analysis, these features are chosen as the principal components of the multiple sequence alignment, which provide a readily accessible low-dimensional representation of dominant sequence variability and can capture underlying structural or functional differences within the protein family. The inference and sampling are performed in an autoregressive manner: coevolutionary data and low-dimensional features are embedded into conditional distributions that are sampled sequentially to guide generation toward specific points or regions in feature space (here, principal component space), thereby enabling a new level of controllability and interpretability in statistical protein design. A conceptually similar approach underlies AF-Cluster [24], which shows that clustering sequences within a single MSA can separate structural signals associated with different conformations. While both approaches exploit this separation, FeatureDCA incorporates it directly into a generative model, enabling continuous conditioning and targeted sequence design.

We demonstrate that FeatureDCA achieves generative accuracy comparable to or better than state-of-the-art models like bmDCA and ArDCA, while also enabling controllable, feature-conditioned sampling. Using PCA-based conditioning, FeatureDCA generates synthetic sequences that accurately reproduce the statistical properties of natural MSAs, such as pairwise correlations and principal component distributions, while maintaining substantial sequence diversity, as measured by Hamming distance. This balance between statistical fidelity and diversity is crucial for effective protein design, where novel sequences must respect evolutionary constraints but also explore unseen regions of sequence space.

In addition to statistical validation and sequence-level metrics, we assess the structural realism of generated sequences using AlphaFold [25,26] and ESMFold [20], showing that FeatureDCA maintains high structural consistency with experimentally determined wild-type folds. As a biologically grounded case study, we focus on the Response Regulator (RR) family (Pfam ID PF00072), a large and diverse set of bacterial proteins involved in signal transduction. RR proteins share a conserved receiver domain but exhibit distinct dimerization architectures depending on their DNA-binding domains. We show that FeatureDCA, when conditioned on PCA-derived features, can selectively generate sequences corresponding to different RR subclasses, illustrating its capacity to navigate biologically meaningful variation and control functional outcomes in sequence design. Finally, we also evaluate the model's predictive power on experimental deep mutational scanning (DMS) data, demonstrating competitive performance in capturing the functional effects of single-residue mutations.

A schematic overview of the FeatureDCA framework, including training on aligned MSAs, conditioning on PCA-derived features, and downstream applications such as conditional sampling and fitness prediction, is provided in Fig 1. Together, our results position FeatureDCA as a powerful and flexible framework for generative modeling in protein sequence space, combining the strength of Potts-based inference with the adaptability of feature-based conditioning.

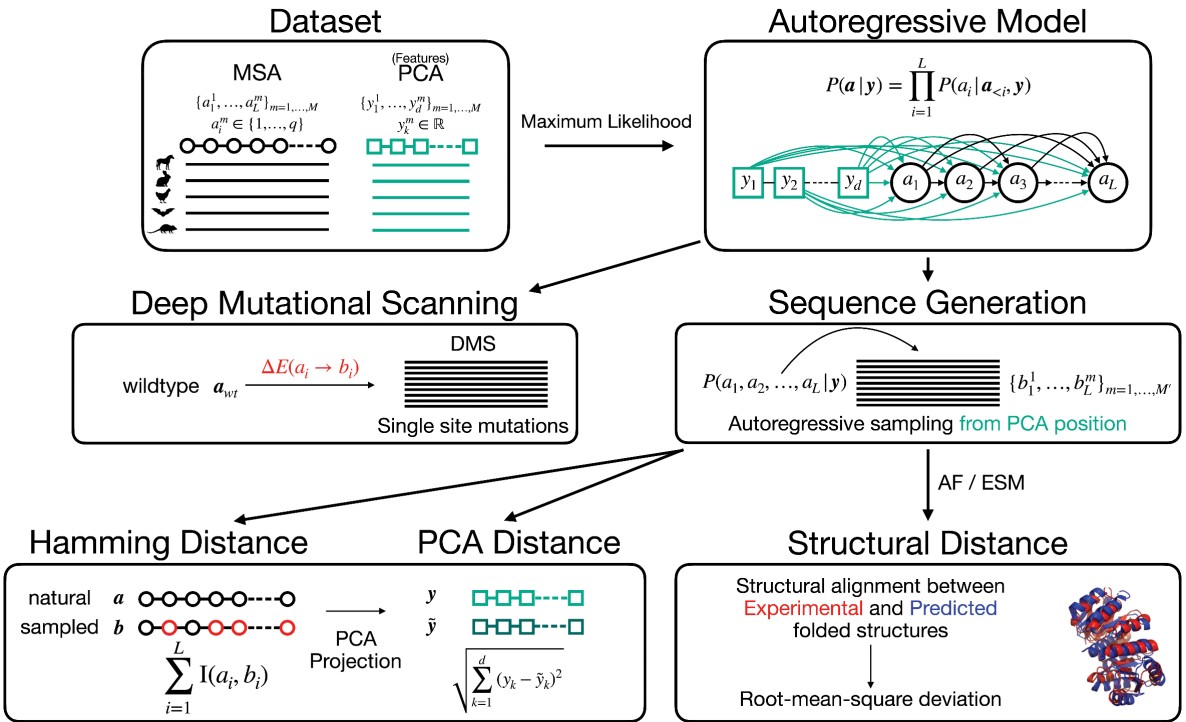

**Fig 1**. **Overview of the generative modeling pipeline**. A multiple sequence alignment (MSA) of homologous sequences is embedded using PCA to produce continuous feature vectors for each sequence. An autoregressive model is trained to learn the conditional distribution $P(a_1, \ldots, a_L | y)$, where $y$ encodes the PCA features. The model is used for multiple downstream tasks: (1) Deep Mutational Scanning (DMS) prediction via single-site substitutions and computation of energy changes $\Delta E(a_i \rightarrow b_i)$; (2) Sequence generation by autoregressive sampling conditioned on $y$; and (3) Evaluation of generated sequences by comparing them to natural sequences using Hamming distance, PCA embedding distance, and structural RMSD between predicted and experimental structures.

## Methods

We present a novel extension of the standard autoregressive Direct Coupling Analysis (ArDCA) method [5], in which sequence-dependent vectors of biologically relevant features are embedded in the amino acid space of a protein family, thereby constraining the sampling of new sequences to specific, user-defined characteristics.

Starting from the exact, autoregressive decomposition of the joint probability distribution

$$P(\mathbf{a}, \mathbf{y}) = P(\mathbf{y}) \prod_{k=1}^{L} P(a_k | \mathbf{a}_{<k}, \mathbf{y}), \tag{1}$$

we model the single-site distribution of amino acid $a_k$, conditioned on amino acids $\mathbf{a}_{<k} = \{a_1, a_2, \ldots, a_{k-1}\}$ and feature vector $\mathbf{y} \in \mathbb{R}^d$ as a Boltzmann measure over the energy function

$$H(a_k | a_{<k}, \mathbf{y}) = -\sum_{i<k} J_{ki}(a_k, a_i) - h_k(a_k) - \mathbf{y} \cdot \mathbf{G}_k(a_k). \tag{2}$$

Here, the first two terms represent the standard Potts model that captures coevolution and conservation signals through an interaction tensor $J_{ki}(a_k, a_i)$ and a local field $h_k(a_k)$ [27,28]. The final term couples the vector of external conditioning characteristics $\mathbf{y}$ with an embedding $\mathbf{G}_k(a_k)$ of the amino acid in the feature space. This formulation effectively

combines evolutionary constraints with additional design conditions, offering a robust, flexible framework for generative protein design. In the present work, each feature vector **y** consists of the first d principal components (PCs) of the corresponding sequence **a**, derived from the multiple sequence alignment and capturing dominant modes of sequence variability that may reflect structural or functional differences within the protein family. This choice is motivated by the need to address sequence generation in specific regions of the PC space, along with the computational simplicity and widespread availability of principal component analysis for multiple sequence alignments. A detailed derivation of the functional form of Eq 2, based on a Gaussian approximation of the conditional distribution, is provided in Appendix A2 in S1 File.

Defined as such, the model is family dependent, with a number of trainable parameters that scales quadratically with the length of the multiple sequence alignment (MSA). Training is performed by an *L2*-regularized maximum likelihood estimation over a joint dataset comprising an MSA and its corresponding conditioning features, here represented by the principal components:

$$\mathcal{D} = \left\{a_i^m\right\}_{i=1,\ldots,L}^{m=1,\ldots,M} \cup \left\{y_\alpha^m\right\}_{\alpha=1,\ldots,d}^{m=1,\ldots,M} \qquad (3)$$

with $a_i^m \in \{1, \ldots, q\}$, $y_\alpha^m \in \mathbb{R}$, where L is the length of the multiple sequence alignment, M the number of sequences in the MSA, d is the dimension of the feature vector, and q is the alphabet size of MSA ($q = 21$ for proteins: 20 amino acids plus the gap symbol, see Appendices A1 and A3 in S1 File for implementation details). Given the $L(L-1)q^2/2 + L(q + dq)$ parameters of the model, the sampling is performed in an autoregressive fashion, following the decomposition in Eq 1. Starting from a given feature vector, amino acids are sampled iteratively, conditioned on the previous positions along the chain. This is a highly efficient sampling method, as it consists of L different samples of univariate distributions of the form:

$$P(a_k|\mathbf{a}_{<k}, \boldsymbol{y}) = \frac{e^{\sum_{i<k} J_{ki}(a_k,a_i)+h_k(a_k)+\boldsymbol{y}\cdot\boldsymbol{G}_k(a_k)}}{\sum_{c=1}^{q} e^{\sum_{i<k} J_{ki}(c,a_i)+h_k(c)+\boldsymbol{y}\cdot\boldsymbol{G}_k(c)}}. \qquad (4)$$

As discussed above, in the present analysis we focus on feature vectors derived from principal component analysis of the multiple sequence alignment. However, the nature of the feature vector can be heterogeneous: it can represent different biological traits, such as structural or functional characteristics, as well as evolutionary or experimental ones. Examples of these can be optimal growth temperatures, folding temperatures, binding specificity, phylogenetic profiles, stability, or mutational fitness.

## Results

### Generativity

The minimal requisite of the model is to sample sequences that are statistically indistinguishable from the natural ones. This means reproducing both the pairwise frequency statistics and the PCA projection of the natural MSA. As a metric for the pairwise frequency statistics, we use the Pearson coefficient of the connected correlations of the natural and generated MSAs:

$$C_{ij}(a_i, a_j) = f_{ij}(a_i, a_j) - f_i(a_i)f_j(a_j), \qquad (5)$$

where $f_i$ and $f_{ij}$ are the single- and pair-wise frequency counts of the alignment, see Appendix A1 in S1 File for details. To quantitatively compare the similarities between the PC projections of the natural and generated datasets, we introduce the entropic-regularized Wasserstein distance, also known as Sinkhorn divergence or Earth Mover's distance [29]. This metric defines the optimal transport distance between two distributions. Technical details and definitions can be found in Appendix A5 in S1 File. As a general benchmark for the generative capabilities of the model, we use bmDCA and standard ArDCA, which are considered the state-of-the-art architectures in terms of accuracy and computational efficiency, respectively. Relative to protein family PF13354 (Beta-lactamase), Fig 2A represents the PCA distribution of the natural MSA compared to those of the MSAs sampled by bmDCA, ArDCA, and FeatureDCA. In particular, each sequence

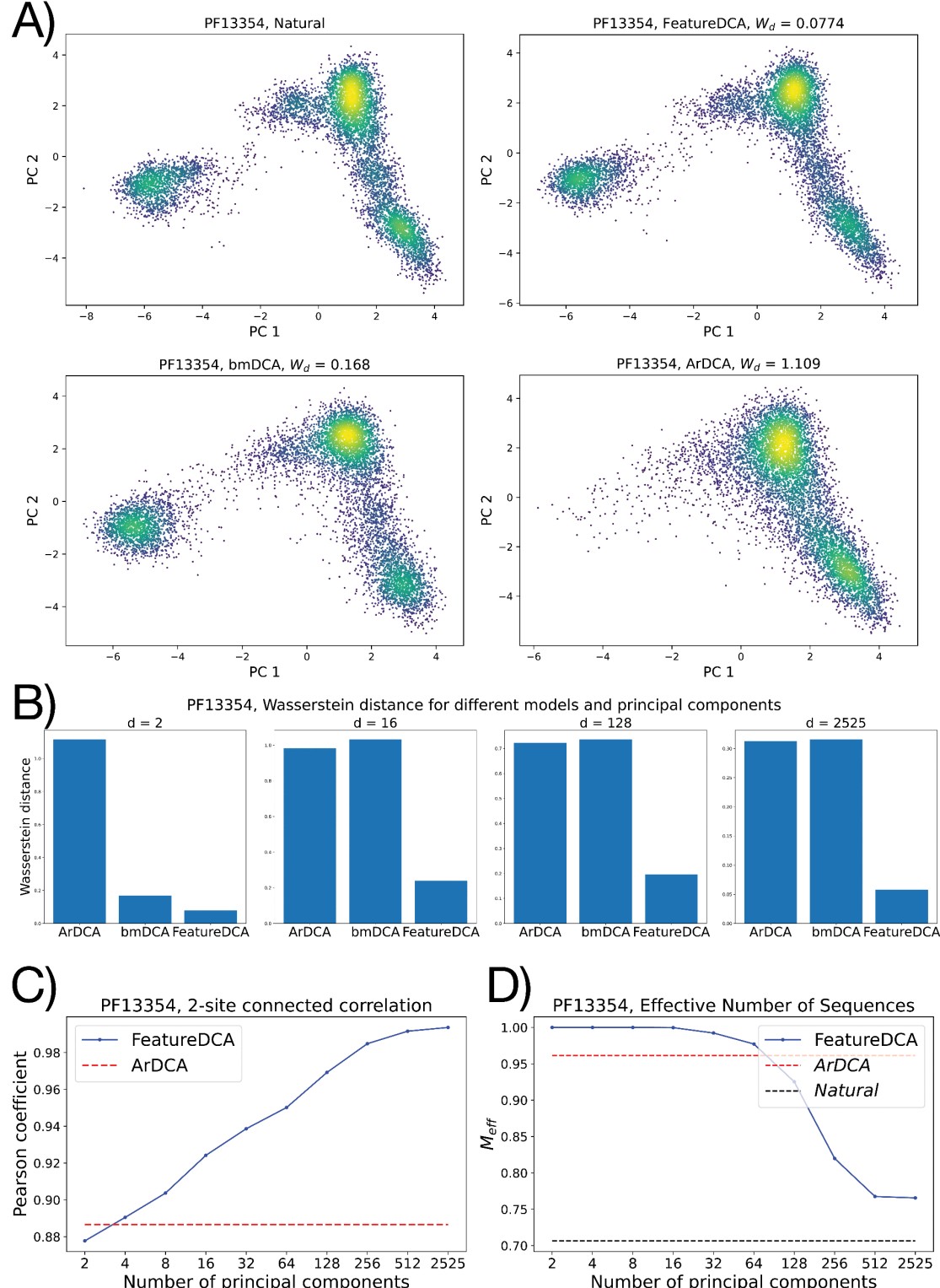

**Fig 2**. **A) Projection onto the first two principal components of natural and generated MSAs from protein family PF13354. B)** Wasserstein distance between the *d*-dimensional PCA distributions of the natural and generated MSA for different models. FeatureDCA was trained with the corresponding number of principal components. **C)** Pearson coefficient of the 2-site connected correlation between natural and generated data as a function of the number of principal components learned during training. **D)** Effective depth percentage of the generated dataset as a function of the number of principal components learned during training.

from FeatureDCA is generated conditioning on a unique $d$-dimensional point in the PCA projection of the natural distribution. In Fig 2A, FeatureDCA is trained with $d = 2$ principal components, which is enough to reproduce a PCA distribution qualitatively and quantitatively indistinguishable from that of bmDCA.

Fig 2B shows the Wasserstein distance between the natural and generated datasets when projected in the $d$-dimensional PCA space for different generative models. In particular, the PCA distribution of data generated from FeatureDCA can accurately reproduce the first $d$ components, provided they have been learned during training, while bmDCA and ArDCA do not reproduce components higher than the second, see Appendix A6 in S1 File. Fig 2C represents the Pearson coefficient of the connected correlation defined in Eq (5) between natural and generated data. As the number of principal components increases, so does the Pearson coefficient relative to FeatureDCA, outperforming either ArDCA or bmDCA. However, when the number of principal components approaches $d_{max} = 2525$, representing the number of components required to explain 99% of the variance of the MSA, the model overfits the data and generates sequences that are identical to the natural ones. In this regime, the model begins to memorize natural sequences rather than generalizing, as indicated by near-zero Hamming distances to training data. Finally, Fig 2D shows the behavior of the effective depth $M_{eff}$, the number of effectively unique sequences as defined in Appendix A1 in S1 File, as a function of the number of principal components used during training.

### In-place generativity

Although the model can globally reproduce the PC projection to the point of improving the generativity of the most reliable models, such as bmDCA, the actual goal of the architecture is to sample sequences at specific points in the PC space. Given a wild-type sequence $\boldsymbol{a}^{wt}$ located at point $\boldsymbol{y}^{wt} \in \mathbb{R}^d$ in the PC space, sequences generated conditioned on that point should be comparable to the wild type in terms of Hamming distance and Euclidean distance in the PC space. In Fig 3, we compare the average Hamming and Euclidean distances of samples of a thousand sequences generated conditionally to three different wild-type sequences as a function of the number of principal components learned during training and used for the sampling. It can be seen that, using fewer principal components, the generated sequences are spread on average over a cloud centered around the corresponding wild type, with a Hamming distance between 55% and 72%. Notably, these distances are comparable to the saturation regime identified by Sanders and Schneider [30], who showed that for long protein chains the Hamming distance between unrelated sequences approaches a value of approximately 75%, whereas homologous sequences are expected to remain below this random saturation limit. As the number of principal components increases, the average position of each generated sample cluster shifts toward its target wild type, while the variance of the distribution decreases. Accordingly, the average Hamming distance between samples and non-corresponding wild types approximates the actual Hamming distance between different wild types. However, when the number of principal components approaches $d_{max}$, the Hamming distance between a sample and its corresponding wild type goes to zero, highlighting the memorization of the natural dataset due to overfitting. From a statistical point of view, a number of principal components between 32 and 128 results in a good trade-off between generative accuracy and generalization.

Using structure prediction models such as AlphaFold 3 (AF) [26] or ESMFold (ESM) [20], we can evaluate the folding of generated sequences as a function of the number of principal components. Starting from two wild-type sequences with experimentally determined crystal structures in the Protein Data Bank (PDB) [31], we fold the generated sequences using ESM and compare their predicted structures to the experimental ones via root-mean-square deviation (RMSD) of atomic positions. However, this analysis is inherently limited by the accuracy of ESM or AF in distinguishing structurally diverse members within the same protein family. In some families, reliable evaluation is not possible, as both ESM and AF fail to capture distinct foldings among naturally aligned but structurally divergent sequences. Panels in Fig 4 illustrate two contrasting cases for protein families PF00014 and PF00076. In the first case (PF00014), the RMSD between the predicted structure of the two wild-type sequences (blue dotted line) does not match the RMSD between their

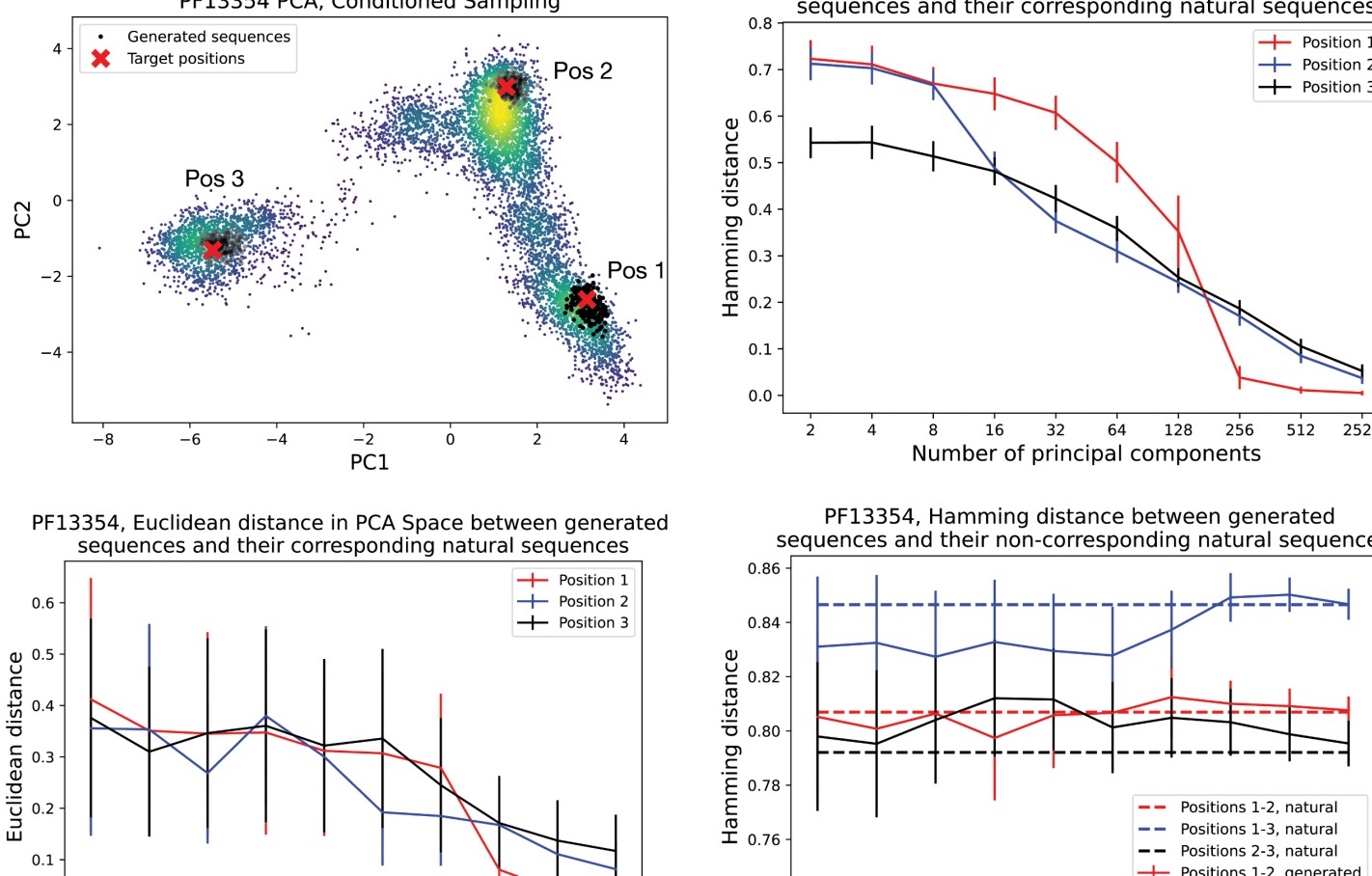

**Fig 3**. **Statistical analysis of the generativity conditioned on three different positions on the PC space as a function of the number of principal components learned during training. Top left**: red crosses represent the three positions chosen on different islands of the PC projection to study the different conditioned sampling. The clouds of black dots represent the generated sequences around the target positions. **Bottom left**: Euclidean distance in the first two PC plane between the target positions and the generated sequences conditioned on those positions. **Top right**: Hamming distance between the target positions and the generated sequences conditioned on those positions. **Bottom right**: Hamming distance between the generated sequences and the non-corresponding target positions.

experimental structures (red dotted line). As a result, the sequences generated with FeatureDCA from either wild type are both folded into the structure of wild type 1. A more detailed structural analysis supporting this interpretation is provided in Appendix A7 in S1 File, where we show, using TM-score and contact-based comparisons, that the discrepancy between predicted and experimental structures reflects a limitation of current structure prediction methods (AF/ESM), rather than a limitation of the RMSD metric itself. In contrast, for PF00076, the structural predictions of ESM closely match the experimental structures, effectively distinguishing between the two conformations. In this case, the sequences generated with FeatureDCA fold into three-dimensional structures that are closely aligned with their respective wild types.

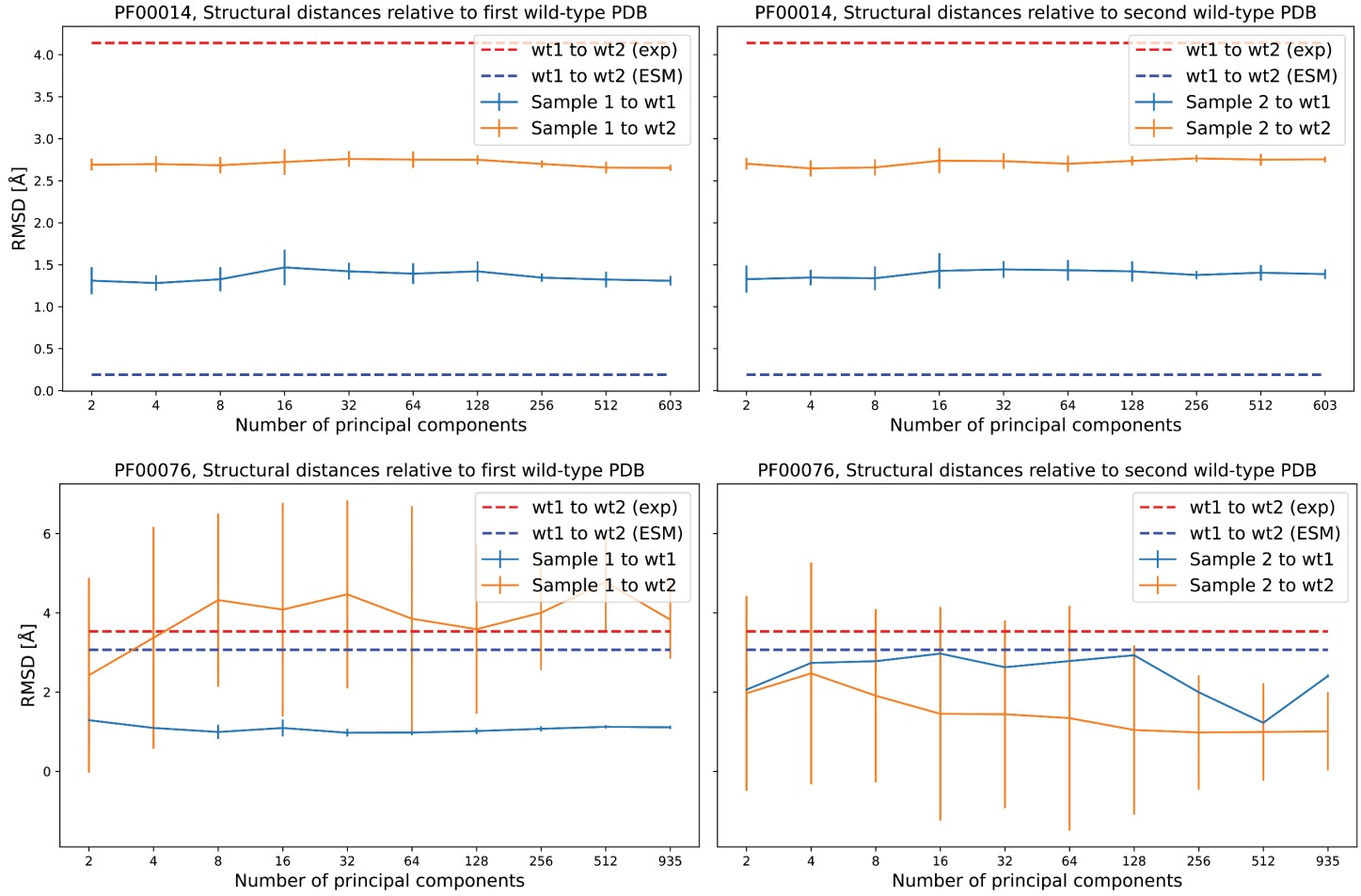

**Fig 4**. **Structural similarity of generated sequences to wild-type structures in protein families PF00014 and PF00076. Each panel shows the RMSD (Å) between the predicted structures of generated sequences and their respective wild-type references, plotted against the number of principal components used for learning and sequence generation. Top**: PF00014; **Bottom**: PF00076. **Left**: RMSD relative to the first wild-type (wt1) structure; **Right**: RMSD relative to the second wild-type (wt2) structure. Dashed lines indicate the RMSD between the two wild types using experimental structures (red) and ESM predictions (blue). Solid lines represent average RMSDs of generated samples to wt1 (blue) and to wt2 (orange), with error bars showing standard deviation. This analysis highlights the structural consistency of sampled sequences relative to the wild types and the capacity of ESM to recover alternative foldings, which appears limited in PF00014 but is more variable in PF00076.

### The case of RR homodimers

The generative capabilities of FeatureDCA can be evaluated on a protein domain characterized by both a structurally diverse landscape and a rich sequence repertoire, ensuring that accurate folding predictions can be obtained across its variants using models such as AlphaFold (AF) or ESMFold (ESM). A prototypical example is the bacterial Response Regulator (RR) family (Pfam ID PF00072), which functions as part of a two-component signal transduction system. In this system, a sensor histidine kinase (HK) detects environmental cues and undergoes autophosphorylation, subsequently transferring the phosphate group to its RR partner. Phosphorylation activates the RR, typically inducing a conformational change that alters its function, most commonly by promoting DNA binding and transcriptional regulation. RRs exhibit architectural diversity depending on their DNA-binding domains, which in turn influence their dimerization mechanisms [32–34]. Among the PF00072 sequences, we focus on three well-characterized subclasses that form distinct homodimers

upon phosphorylation. These dimerization modes correspond to distinct domain architectures, each comprising the conserved receiver domain PF00072 combined with one of three DNA-binding domains: Trans_Reg_C (PF00486), GerE (PF00196), or LytTR (PF04397). Fig 5 shows representative homodimer structures from the three RR subclasses (A–C) and their pairwise structural alignments (D–F), performed using PyMOL's `align` function [35], which reports the root-mean-square deviations (RMSDs) between matched atoms. The PDB IDs corresponding to the Trans_Reg_C, LytTR, and GerE classes are 1NXS, 4CBV, and 4ZMS, respectively. The RMSDs for the three pairwise alignments are: 8.6 Å between Trans_Reg_C and LytTR (D), 16.1 Å between Trans_Reg_C and GerE (E), and 18.7 Å between LytTR and GerE (F).

To evaluate the generative capabilities of FeatureDCA, which learns from and generates aligned sequences, it is essential to verify that AF or ESM can accurately fold natural sequences from the alignment into the correct homodimer classes observed in experimental structures. Among the 1.7 million UniProt sequences assigned to PF00072, we identified 160,585 with an architecture compatible with the Trans_Reg_C class, 69,401 with GerE, and 34,699 with LytTR, for a total of 264,685 non-redundant sequences. Based on these, we constructed a multiple sequence alignment of 118 positions, longer than the 111-position Hidden Markov Model (HMM) profile available in Pfam (now integrated into InterPro) [36,37], ensuring coverage of the relevant structural features. When folded with AlphaFold 3, the aligned natural sequences adopt structures highly consistent with their experimental counterparts: the root-mean-square deviations between predicted and crystal structures are 0.35 Å for 1NXS (Trans_Reg_C), 1.1 Å for 4CBV (LytTR), and 1.3 Å for 4ZMS (GerE). These values fall within or below the resolution range of the experimental structures (1.5–4 Å), confirming the structural compatibility of the alignment for downstream generative modeling. An interesting feature of the dataset

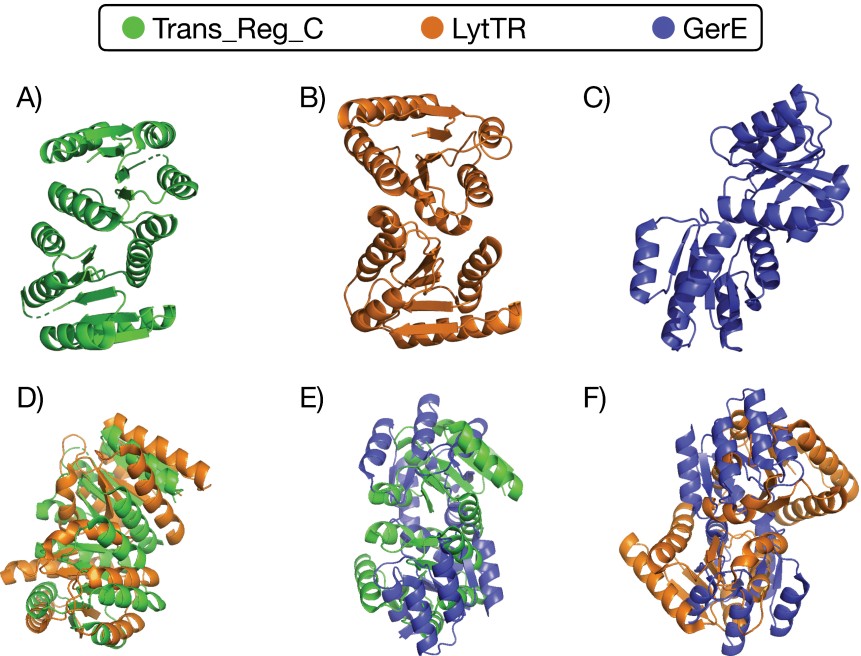

**Fig 5**. **Structural comparison of three Response Regulator (RR) subclasses with distinct dimerization modes**. **A–C)** Experimentally determined homodimer structures of representative RR proteins containing the DNA-binding domains Trans_Reg_C (A, PDB ID: 1NXS), LytTR (B, PDB ID: 4CBV), and GerE (C, PDB ID: 4ZMS). **D–F)** Pairwise structural alignments between these dimers, performed using PyMOL's `align` function. The root-mean-square deviations (RMSDs) between the dimers quantify the extent of structural divergence: 8.6 Å between Trans_Reg_C and LytTR (D), 16.1 Å between Trans_Reg_C and GerE (E), and 18.7 Å between LytTR and GerE (F). These values highlight the substantial differences in quaternary structure and dimerization geometry across the three RR subclasses, despite all sharing the same conserved receiver domain fold.

is that the three subsets corresponding to the homodimer classes form well-separated clusters when projected onto the first two principal components. This clear separation in sequence space, as shown in Fig 6, supports the idea that FeatureDCA can learn to condition sequence generation on structural class.

To assess this claim, we performed a targeted sampling experiment. For each structural class, we extracted the principal component (PC) coordinates of the sequences corresponding to the experimental structures (1NXS, 4ZMS, or 4CBV) and used them as conditioning inputs to generate new sequences. These generated sequences were then folded using AlphaFold 3 and compared, via RMSD of the aligned atomic positions, to all three experimental wild-type structures. The top-right, top-left, and bottom-left panels in Fig 7 show the results of this experiment. Each panel displays the average RMSD between the predicted structures of generated sequences and all three wild-type references, plotted as a function of the number of principal components used during model training and generation. The RMSD to the structure of the same class used for conditioning is expected to decrease with increasing number of PCs, while the RMSDs to the other two reference structures tend to approach the experimentally measured distances between the corresponding wild-type structures (dotted lines).

The trend shown in Fig 7 confirms that FeatureDCA can effectively guide generation toward the intended structural class, provided that sufficient principal component information is used during training. However, there are clear differences in class-specific performance. The Trans_Reg_C class (1NXS), which is the most represented in the dataset, is consistently reproduced with the lowest RMSDs, indicating that FeatureDCA learns its statistical and structural features more effectively. In contrast, the LytTR class (4CBV) presents a greater challenge: for low-dimensional conditioning, its generated sequences often adopt structures more similar to Trans_Reg_C than to LytTR itself, suggesting confusion between classes. Only when the number of PCs becomes sufficiently large does the model begin to distinguish LytTR with sufficient resolution. These observations suggest that both training-set representation and feature dimensionality play key roles in enabling class-specific generative control. Finally, the bottom-right panel in Fig 7 quantifies the success rate

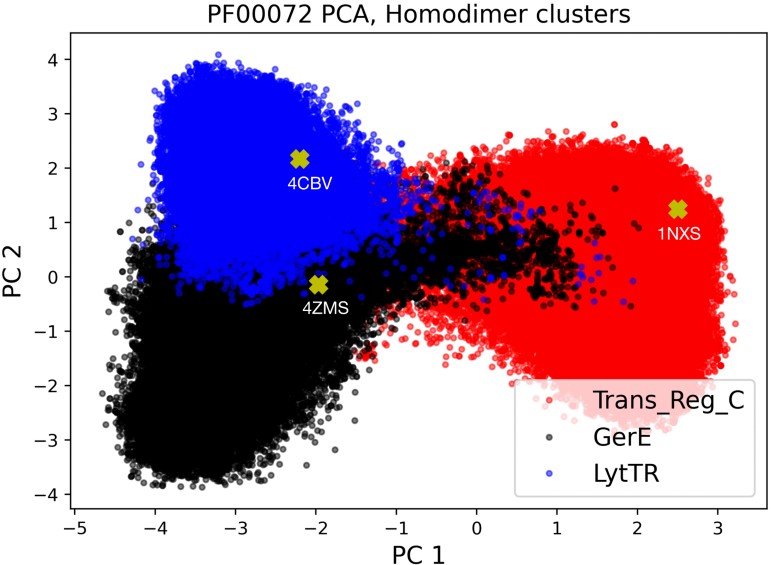

**Fig 6**. **PCA projection of aligned RR sequences reveals distinct structural clusters**. Projection of the multiple sequence alignment of PF00072 sequences onto the first two principal components. The three main clusters correspond to sequences containing the Trans_Reg_C (red dots), LytTR (blue), and GerE (black) DNA-binding domains, associated with distinct homodimerization classes. Yellow crosses mark the positions of the experimental PDB structures 1NXS (Trans_Reg_C), 4CBV (LytTR), and 4ZMS (GerE), each falling within its respective cluster. This separation supports the idea that structural class is encoded in sequence space and can be learned by FeatureDCA.

RMSD Analysis Between AlphaFold-Predicted and Experimental Structures for Generated Sequences in PF00072 with FeatureDCA

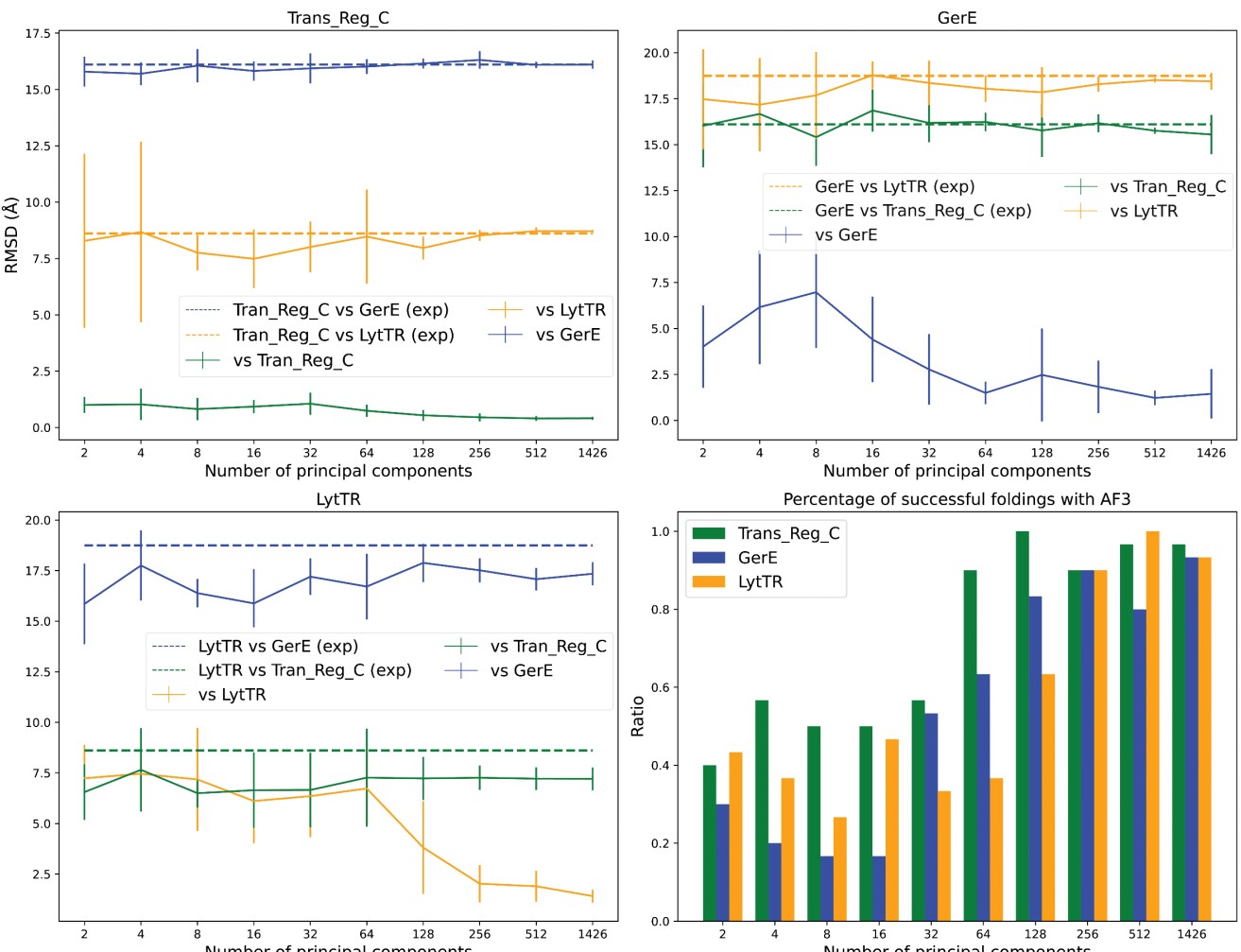

**Fig 7**. **Structural consistency and class specificity of FeatureDCA-generated sequences for the PF00072 protein family**. The **top-left**, **top-right**, and **bottom-left** panels show the root-mean-square deviation (RMSD) between predicted structures of generated sequences from one of the three RR structural classes, Trans_Reg_C (PDB ID: 1NXS), GerE (PDB ID: 4ZMS), and LytTR (PDB ID: 4CBV), respectively, and the three experimental wild-type structures. For each class, sequences were generated by conditioning FeatureDCA on the principal component (PC) coordinates of the sequences corresponding to the experimental PDB structures, and the predicted structures of these generated sequences were compared to all three references using AlphaFold 3. Solid lines represent the average RMSD as a function of the number of PCs used during training and generation. In each plot, the curve corresponding to the reference structure of the same class (e.g., 1NXS for Trans_Reg_C, green curve in the top right panel) is expected to show the lowest RMSD, while the other two curves converge toward the RMSDs between the experimental structures (shown as dotted lines), reflecting the divergence across classes. The **bottom-right** panel shows the fraction of generated sequences whose predicted structures are within 10 Å RMSD of the corresponding class-specific wild-type structure, indicating successful fold recovery. This analysis demonstrates that FeatureDCA not only generates structurally plausible sequences, but also preserves class-specific structural identity as a function of PC-based conditioning.

of folding the generated sequences with AlphaFold. A fold is considered successful if its predicted structure lies within a threshold RMSD to the corresponding target structure; here, the threshold is set to 10 Å. Alternatively, one could assess folding quality using the predicted Local Distance Difference Test (pLDDT) score returned by AlphaFold. This analysis, reported in Appendix A8 in S1 File, supports the same qualitative conclusions: sequences conditioned on well-sampled structural classes are more likely to produce high-confidence, accurate folds.

The results confirm that FeatureDCA, when appropriately conditioned, can generate structurally plausible sequences that reflect subclass-specific geometries, but also highlight the limits of low-dimensional conditioning in sparsely represented regions of sequence space.

## Predicting mutational effects via in-silico deep mutational scanning

An important benchmark for energy-based models of protein sequences is their ability to capture the beneficial, neutral, or deleterious effects of mutations. In particular, in-silico deep mutational scanning (DMS) provides a standard way to evaluate statistical models trained on multiple sequence alignments by comparing predicted mutational scores to experimentally measured fitness effects. It is important to note that experimental DMS measurements typically report a composite fitness signal, which may arise from changes in thermodynamic stability, functional activity, or other biophysical constraints, and do not generally disentangle these contributions. Similarly, DCA-based models do not explicitly distinguish between different physical origins of mutational effects, but instead assign mutations a score based on changes in sequence likelihood, which has been shown to correlate with experimental DMS measurements [38]. For the model presented here, the mutational score can be defined as:

$$\Delta E(a_i \rightarrow b_i) = -\log \frac{P(a_1, ..., a_{i-1}, b_i, a_{i+1}, ..., a_L, \tilde{y})}{P(a_1, ..., a_{i-1}, a_i, a_{i+1}, ..., a_L, y)}, \tag{6}$$

where $P(a, y)$ represents the joint probability distribution of the autoregressive model defined by Eq (1, 4), while $y$ and $\tilde{y}$ are the PC projections of the wild-type and mutant sequence, respectively.

Following the approach of Trinquier et al. [5], we benchmarked FeatureDCA against ArDCA on the well-studied TEM-1 Beta-lactamase family (PF13354), for which extensive experimental DMS data are available [8,39,40]. In particular, we use deep mutational scanning data from Ostermeier and collaborators [40] consisting of all single-amino-acid substitutions with available experimental fitness scores. Additional details on data curation and links to the relevant data repositories are provided in Appendix A9 in S1 File.

Model performance was evaluated using the Spearman rank correlation between the predicted and experimental mutational effects. As shown in Fig 8, for small conditioning dimensions $d$, FeatureDCA achieves performance comparable to ArDCA, with Spearman correlations of approximately $\rho \approx 0.70$ for both models. Interestingly, as the number of principal components used during training increases, we observe a non-monotonic behavior in predictive performance. Specifically, the correlation initially decreases, suggesting that conditioning on an excessive number of PCs may overfit or dilute mutational signals. However, at larger values of $d$, the performance begins to recover, indicating that the model eventually re-stabilizes and incorporates additional global variation effectively.

These results demonstrate that FeatureDCA not only supports conditional generative modeling but also remains competitive in mutational effect prediction, with performance that can be tuned via the dimensionality of the conditioning space.

## Discussion

In this work, we presented FeatureDCA, a feature-conditioned extension of the Direct Coupling Analysis (DCA) framework, designed for controllable generation of protein sequences within a given protein family. By integrating biologically relevant, low-dimensional features, specifically the principal components (PCs) derived from multiple sequence alignments (MSAs), FeatureDCA enables users to direct sequence generation toward targeted regions of sequence space, all while preserving the statistical and structural characteristics of natural proteins used for training.

Through extensive simulations, we showed that FeatureDCA accurately reproduces key aspects of protein families, including coevolutionary statistics, principal component distributions, and structural properties. Notably, the model

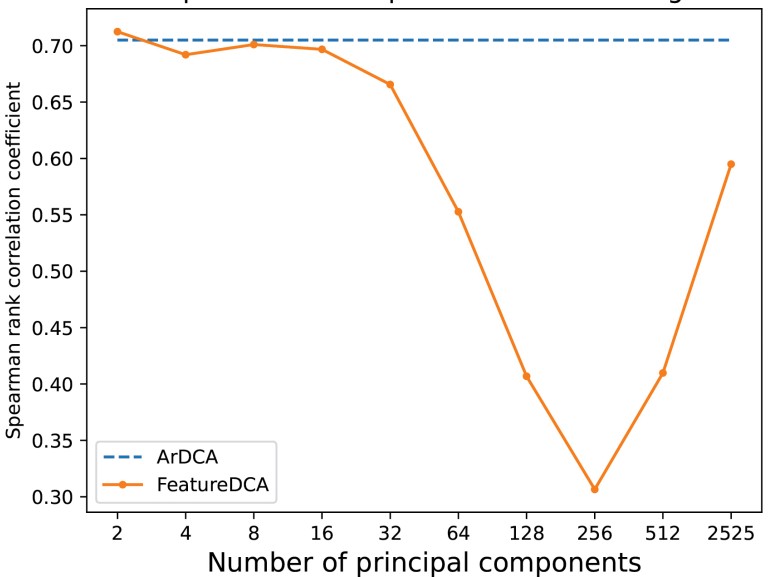

**Fig 8**. **Predictive performance of FeatureDCA and ArDCA on experimental deep mutational scanning data for Beta-lactamase (PF13354)**. The Spearman rank correlation coefficient between predicted and experimental mutational effects is shown as a function of the number of principal components used during training and generation. ArDCA corresponds to the unconditioned autoregressive baseline. FeatureDCA performs comparably to ArDCA for small values of the number of principal components $d$, but displays a non-monotonic behavior: the correlation initially decreases as more components are introduced, then increases again at larger $d$, suggesting a complex trade-off between dimensionality, signal strength, and generalization.

generates novel sequences that remain at a substantial Hamming distance from training examples, ensuring diversity, an essential criterion in protein design.

When compared to established models such as bmDCA and ArDCA, FeatureDCA delivers competitive or superior generative performance and introduces a crucial new capability: directional control over the generated sequence distribution, as illustrated in Fig 2. The growing interest in controllable generative models is underscored by a very recent preprint that, while we were finalizing this manuscript, introduced a complementary approach: a data-augmentation pipeline that combines protein language model embeddings with label-informed RBMs to enable label-specific generation of homologous protein sequences [41]. In contrast to our effort to extend controllability using continuous, unlabeled, biologically interpretable features, their approach leverages augmented categorical labels to steer generation. Together, these complementary strategies highlight the increasing focus on controllability in protein design.

Building on this theme, we showed that FeatureDCA enables not only global steering along principal-components, but also localized generation around user-defined points in feature space. This capability allows the model to sample sequences that approximate specific wild-type sequences both statistically and structurally, see Figs 3 and 4. In particular, the case study on the Response Regulator (RR) family (PF00072) highlighted FeatureDCA's capacity to navigate intrafamily structural diversity, similarly to what is shown in [24]. By conditioning on PC coordinates of known RR subtypes, shown in Figs 5 and 6, the model was able to selectively generate sequences that folded into class-specific dimerization geometries, demonstrating its potential for targeted generative design. In addition, we showed that FeatureDCA retains predictive power on experimental deep mutational scanning data, further confirming its ability to capture local functional constraints in sequence space.

Despite these advances, limitations remain. First, the conditioning space is currently restricted to principal components that, while biologically meaningful, are not explicitly linked to function or structure. Extending FeatureDCA to condition on experimentally or computationally derived features, such as binding specificity, fitness, or structural class, would broaden its applicability without changing the architecture of the outlined model. Second, generative performance in sparsely populated regions of feature space, such as the LytTR subclass for the case of RR domains, is limited, reflecting the challenge of learning from underrepresented sequence types. More balanced or augmented training sets may help alleviate this problem.

A key strength of FeatureDCA lies in its flexibility. Unlike standard Potts-based models that rely on global sampling schemes, the autoregressive implementation used to solve the model allows for efficient training and generation by factorizing the sequence distribution into a chain of conditionals. This not only makes FeatureDCA computationally comparable to ArDCA but also allows the sampling process to be explicitly guided by external features beyond local amino acid frequencies. As demonstrated in this work, such conditioning enables both global and localized control over the generative process, opening the door to a broad range of applications in statistical modeling and protein design.

In summary, FeatureDCA bridges the gap between interpretable statistical models and flexible, feature-guided protein design. It provides a robust and efficient framework for controllable sequence generation grounded in evolutionary data, with the potential to guide functional exploration, structural modeling, and synthetic biology applications.

## Supporting information

**S1 File. A file that contains all the supporting information for the paper.** Including: Appendix A1: Direct Coupling Analysis Details; Appendix A2: Mathematical foundation of the feature-conditioned autoregressive model; Appendix A3: Implementation details; Appendix A4: data processing and MSA construction; Appendix A5: Wasserstein distance and Sinkhorn divergence; Appendix A6: Principal components higher than the second; Appendix A7: Additional structural analysis of the PF00014 mismatch case; Appendix A8: pLDDT analysis for generated RR homodimer sequences; Appendix A9: In-silico deep mutational scanning; Appendix A10: Supplementary figures.
(PDF)

## Acknowledgments

We are deeply grateful to Martin Weigt and Leonardo Di Bari for many interesting discussions on addressable sequence generation.

## Author contributions

**Conceptualization:** Paolo De Los Rios, Andrea Pagnani.

**Data curation:** Francesco Caredda, Lisa Gennai.

**Formal analysis:** Francesco Caredda.

**Methodology:** Francesco Caredda, Paolo De Los Rios, Andrea Pagnani.

**Supervision:** Paolo De Los Rios, Andrea Pagnani.

**Visualization:** Francesco Caredda.

**Writing – original draft:** Francesco Caredda.

**Writing – review & editing:** Francesco Caredda, Lisa Gennai, Paolo De Los Rios, Andrea Pagnani.

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
