## [Decision Letter · Decision Letter 0]

16 Dec 2025

PCOMPBIOL-D-25-01907

Controllable protein design via autoregressive Direct Coupling Analysis conditioned on principal components

PLOS Computational Biology

Dear Dr. CAREDDA,

Thank you for submitting your manuscript to PLOS Computational Biology. After careful consideration, we feel that it has merit but does not fully meet PLOS Computational Biology's publication criteria as it currently stands. Therefore, we invite you to submit a revised version of the manuscript that addresses the points raised during the review process.

We look forward to receiving your revised manuscript.

Kind regards,

Fei Guo

Academic Editor

PLOS Computational Biology

Ferhat Ay

Section Editor

PLOS Computational Biology

**Journal Requirements:**

1) Your manuscript is missing the following sections: Methods.  Please ensure all required sections are present and in the correct order. Make sure section heading levels are clearly indicated in the manuscript text, and limit sub-sections to 3 heading levels. An outline of the required sections can be consulted in our submission guidelines here:

4) Please amend your detailed Financial Disclosure statement. This is published with the article. It must therefore be completed in full sentences and contain the exact wording you wish to be published.

**Reviewers' comments:**

Reviewer's Responses to Questions

**Comments to the Authors:**

Reviewer #1: The manuscript describes an extension of the standard DCA, routinely used in connection with structure prediction of proteins, to include annotated functional data. In my opinion, this is a very relevant study that further improves a technique which has already proven seminal in the field of computational biology. Therefore, I recommend its publication with minor revision, mainly focused on the clarity of the text.

1. "Lack of control over the output distribution" (line 26) seems really pessimistic; maybe it is not exactly what the authors meant to say.

2. Little is said in the main text of the functional features that are included in the model (the y_k quantities). It could be beneficial to the reader to have a clear idea of what they are from the very beginning.

3. As usual, the gaps are treated as a type of amino acid, but I would find it strange they contribute actively to defining function, as eq. 2 would suggest. Do they obey some special rule?

4. A classic work by Sanders and Schneider (Proteins Structure. Funct. Gen. 9, 56, 1991) defines a curve in the space of hamming distance/sequence length that separates homologs from pairs with unknown relation, saturating at distance 0.75 for long chains. Can the maximum Hamming distance found in the generated sequence (line 179) related to that?

5. Predictions are compare with experimental structure with a RMSD (line 200), but it is well known that RMSD displays very nonlinear effects, structural variations producing large changes in the RMSD when this is small and negligible changes when it is large. Can the mismatch observed in the case of PF00014 be analysed in terms of number of common contacts, defined in some appropriate way?

6. I would have preferred to see some results concerning mutations (line 288) in the main text. Is it possible to distinguish between effects of mutations on thermodynamic stability and on function (I.e., all the rest)?

Reviewer #2: Minor Comments

Overall, the manuscript presents an interesting approach for conditioning autoregressive protein sequence models on PCA-derived features. The method is clearly explained at a high level, and the comparisons against existing DCA-based models are valuable. I have several minor comments that, if addressed, would improve clarity, reproducibility, and contextual grounding of the work.

1. Clarification of model architecture

The manuscript refers to the use of an autoregressive model but does not describe the underlying architecture. Please specify whether the model is transformer-based, attention-based, convolutional, or uses another design, and include the total number of parameters or layers.

2. Dataset sizes and splitting strategy

The authors should provide explicit details on the size of the training MSAs (number of sequences, average length), the size of the evaluation datasets, and the procedure used to split or select data for training and evaluation. Clarifying whether they rely on standard Pfam splits or their own selection criteria would improve experimental transparency.

3. Discussion of recent MSA-based generative models

The introduction would benefit from situating the method within the context of more recent MSA-based generative models such as PoET-2 and Tranception, in addition to bmDCA and ArDCA. A brief comparison or rationale for focusing on DCA-based baselines would help readers understand the positioning of this work.

4. Structural prediction confidence metrics

The structural evaluation relies on RMSD comparisons between predicted and experimental structures. It would be helpful to also report pLDDT or other confidence metrics from AlphaFold 3 or ESMFold when folding generated sequences.

5. Amino acid composition analysis

To further validate the generative capabilities of the model, the authors could provide amino acid frequency distributions for natural versus generated sequences. This would allow readers to assess whether the model captures basic sequence-level statistics in addition to PCA and Hamming distance metrics.

6. Details of DMS evaluation

For the DMS experiments, please report the number of mutations or samples included and provide the performance metrics of each model (FeatureDCA, bmDCA, ArDCA) directly in the main text. This would help clarify the extent and rigor of the DMS evaluation.

**Have the authors made all data and (if applicable) computational code underlying the findings in their manuscript fully available?**

Reviewer #1: Yes

Reviewer #2: Yes

PLOS authors have the option to publish the peer review history of their article (what does this mean?). If published, this will include your full peer review and any attached files.

Reviewer #1: No

Reviewer #2: No

**Figure resubmission:**
---

## [Decision Letter · Decision Letter 1]

6 Feb 2026

Dear Dr CAREDDA,

We are pleased to inform you that your manuscript 'Controllable protein design via autoregressive Direct Coupling Analysis conditioned on principal components' has been provisionally accepted for publication in PLOS Computational Biology.

Best regards,

Fei Guo

Academic Editor

PLOS Computational Biology

Ferhat Ay

Section Editor

PLOS Computational Biology

Reviewer's Responses to Questions

**Comments to the Authors:**

Reviewer #1: All my criticisms have been addressed satisfactorily. The result is a convincing and interesting manuscript, that can be published in its present form.

Reviewer #2: Authors have addressed all of my comments!

**Have the authors made all data and (if applicable) computational code underlying the findings in their manuscript fully available?**

Reviewer #1: None

Reviewer #2: None

PLOS authors have the option to publish the peer review history of their article (what does this mean?). If published, this will include your full peer review and any attached files.

Reviewer #1: No

Reviewer #2: No

---

## [Editor Report · Acceptance letter]

PCOMPBIOL-D-25-01907R1

Controllable protein design via autoregressive Direct Coupling Analysis conditioned on principal components

Dear Dr CAREDDA,

I am pleased to inform you that your manuscript has been formally accepted for publication in PLOS Computational Biology. Your manuscript is now with our production department and you will be notified of the publication date in due course.

With kind regards,

Anita Estes
